# REINFORCEMENT LEARNING WITH PROBABILISTICALLY COMPLETE EXPLORATION

## ABSTRACT

Balancing exploration and exploitation remains a key challenge in reinforcement learning (RL). State-of-the-art RL algorithms suffer from high sample complexity, particularly in the sparse reward case, where they can do no better than to explore in all directions until the first positive rewards are found. To mitigate this, we propose Rapidly Randomly-exploring Reinforcement Learning (R3L). We formulate exploration as a search problem and leverage widely-used planning algorithms such as Rapidly-exploring Random Tree (RRT) to find initial solutions. These solutions are used as demonstrations to initialize a policy, then refined by a generic RL algorithm, leading to faster and more stable convergence. We provide theoretical guarantees of R3L exploration finding successful solutions, as well as bounds for its sampling complexity. We experimentally demonstrate the method outperforms classic and intrinsic exploration techniques, requiring only a fraction of exploration samples and achieving better asymptotic performance.

## 1 INTRODUCTION

Reinforcement Learning (RL) studies how agents can learn a desired behaviour by simply using interactions with an environment and a reinforcement signal. Central to RL is the long-standing problem of balancing *exploration* and *exploitation*. Agents must first sufficiently explore the environment to identify high-reward behaviours, before this knowledge can be exploited and refined to maximize long-term rewards. Many recent RL successes have been obtained by relying on well-formed reward signals, that provide rich gradient information to guide policy learning. However, designing such informative rewards is challenging, and rewards are often highly specific to the particular task being solved. Sparse rewards, which carry little or no information besides binary success or failure, are much easier to design. This simplicity comes at a cost; most rewards are identical, so that there is little gradient information to guide policy learning. In this setting, the sample complexity of simple exploration strategies was shown to grow exponentially with state dimension in some cases (Osband et al., 2016b). Intuition behind this phenomenon can be gained by inspecting Figure 1a: exploration in regions where the return surface is flat leads to a random walk type search. This inefficient search continues until non-zero gradients are found, which can then be followed to a local optimum.

Planning algorithms can achieve much better exploration performance than random walk by taking search history into account (Lavalle, 1998). These techniques are also often guaranteed to find a solution in finite time if one exists (Karaman & Frazzoli, 2011). In order to leverage the advantages of these methods, we formulate RL exploration as a planning problem in the state space. Solutions found by search algorithms are then used as demonstrations for RL algorithms, initializing them in regions of policy parameter space where the return surface is not flat. Figure 1b shows the importance of such good initialization; surface gradients can be followed, which greatly facilitates learning.

This paper brings the following contributions. We first formulate RL exploration as a planning problem. This yields a simple and effective method for automatically generating demonstrations without the need for an external expert, solving the planning problem by adapting the classic Rapidly-exploring Random Tree algorithm (RRT) (Kuffner & LaValle, 2000). The demonstrations are then used to initialize an RL policy, which can be refined with a classic RL method such as TRPO (Schulman et al., 2015). We call the proposed method Rapidly Randomly-exploring Reinforcement Learning (R3L)[1], provide theoretical guarantees for finding successful solutions

---

[1]Code will be made available on Github.

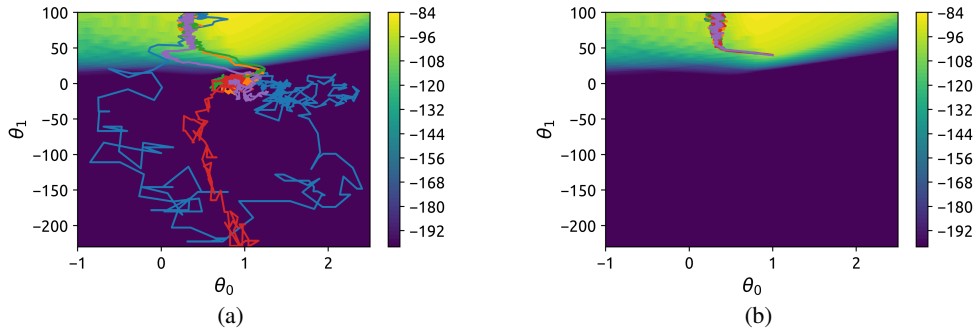

Figure 1: Expected returns achieved by linear policy with 2 parameters on Sparse MountainCar domain (background). Gradient is 0 in the dark blue area. Trajectories show the evolution of policy parameters over 1000 iterations of TRPO, with 5 random seeds. Same colors indicate the same random seeds. (a) Random-walk type behaviour observed when parameters are initialized using Glorot initialization (Glorot & Bengio, 2010). (b) Convergence observed when parameters are initialized in a region with gradients $(1, 40)$.

and derive bounds for its sampling complexity. Experimentally, we demonstrate R3L improves exploration and outperforms classic and recent exploration techniques, and requires only a fraction of the samples while achieving better asymptotic performance. Lastly, we show that R3L lowers the variance of policy gradient methods such as TRPO, and verify that initializing policies in regions with rich gradient information makes them less sensitive to initial conditions and random seed.

The paper is structured as follows: Section 2 analyzes the limitations of classic RL exploration. Section 3 describes R3L and provides theoretical exploration guarantees. Related work is discussed in Section 4, followed by experimental results and comments in Section 5. Finally, Section 6 concludes and gives directions for future work.

## 2 SPARSE-REWARD RL AS RANDOM WALK

Many recent RL methods are based on a policy gradient optimization scheme. This approach optimizes policy parameters $\theta$ with gradient descent, using a loss function $\mathcal{L}(\theta)$ (e.g. expected return) and gradient $g(\theta) \equiv \nabla_\theta \mathcal{L}(\theta)$. Since computing $\mathcal{L}(\theta)$ exactly is intractable, it is common to use unbiased empirical estimators $\hat{L}(\theta)$ and $\hat{g}(\theta)$, estimated from samples acquired by executing the policy. Optimization of $\theta$ then follows the common *stochastic gradient descent* (SGD) update-rule (Bottou, 2010; Robbins & Monro, 1951): $\theta_{n+1} = \theta_n - \epsilon \hat{g}(\theta_n)$, where $\epsilon$ is the learning rate.

The SGD update rule defines a discrete-time stochastic process (Mandt et al., 2017). Note that $\hat{g}$ is the mean of $n_{mb}$ i.i.d. samples. Following the central limit theorem, the distribution over $\hat{g}$ is approximately $\hat{g}(\theta) \sim \mathcal{N}(g(\theta), \frac{1}{n_{mb}}C(\theta))$, meaning $\hat{g}$ is an unbiased estimator of $g$ with covariance $\frac{1}{n_{mb}}C(\theta)$. Consequently, the update rule can be rewritten as (Mandt et al., 2017):

$$\theta_{n+1} = \theta_n - \epsilon g(\theta_n) + \frac{\epsilon}{n_{mb}}B\Delta W, \qquad \Delta W \sim \mathcal{N}(0, \mathbb{I}). \tag{1}$$

Here we assume that $C(\theta) = C$, i.e. approximately constant w.r.t. $\theta$, and factorizes as $C = BB^T$.

SGD is efficient for high-dimensional problems as it offers almost dimension independent convergence rates (Nesterov, 2018). However, SGD requires non-zero gradients to guide the search towards the optimum $\theta^*$, i.e. $|g(\theta)| > \epsilon_g, \forall \theta \neq \theta^*, \epsilon_g \in \mathbb{R}$. In the case of sparse-reward RL problems, such as in Figure 1, much of the loss surface is flat. This leads to inefficient exploration of parameter space $\Theta$, as the drift component in Eq. (1) $g \approx 0$, turning the SGD to a random walk in $\Theta$; $\Delta\theta = \frac{\epsilon}{n_{mb}}B\Delta W$. Random walk is guaranteed to wander to infinity when dimensionality $d_\Theta \geq 3$ (Pólya, 1921; Kakutani, 1944). However, the probability of it reaching a desired region in $\Theta$, e.g. where $g \neq 0$, depends heavily on problem specific parameters. The probability of $\theta_n$ ever reaching a sphere of radius $r$ centered at $\mathcal{C}$ such that $\|\mathcal{C} - \theta_0\| = R > r$ is (Dvoretzky & Erdős, 1951):

$$\Pr\{\|\theta_n - \mathcal{C}\| < r, \text{ for some } n > 0\} = \left(\frac{r}{R}\right)^{d_\Theta - 2} < 1, \qquad \forall d_\Theta \geq 3. \tag{2}$$

In sparse RL problems $r < R$, thus the probability of reaching a desired region by random walk is smaller than 1, i.e. there are no guarantees of finding any solution, even in infinite time. This is in stark contrast with the R3L exploration paradigm, as discussed in Section 3.5.

# 3 R3L: RAPIDLY AND RANDOMLY-EXPLORING REINFORCEMENT LEARNING

R3L adapts RRT to the RL setting by formulating exploration as a planning problem in state space. Unlike random walk, RRT encourages uniform coverage of the search space and is probabilistically complete, i.e. guaranteed to find a solution (Kleinbort et al., 2019).

R3L is decomposed into three main steps: (i) exploration is first achieved using RRT to generate a data-set of successful trajectories, described in Sections 3.2 and 3.3, (ii) successful trajectories are converted to a policy using learning from demonstrations (Section 3.4), and (iii) the policy is refined using classic RL methods.

## 3.1 DEFINITIONS

This work is based on the Markov Decision Process (MDP) framework, defined as a tuple $< \mathcal{S}, \mathcal{A}, T, R, \gamma >$. $\mathcal{S}$ and $\mathcal{A}$ are spaces of states $s$ and actions $a$ respectively. $T : \mathcal{S} \times \mathcal{A} \times \mathcal{S} \rightarrow [0, 1]$ is a transition probability distribution so that $T(s_t, a_t, s_{t+1}) = p(s_{t+1}|s_t, a_t)$, where the subscript $t$ indicates the $t^{th}$ discrete timestep. $R : \mathcal{S} \times \mathcal{A} \times \mathcal{S} \rightarrow \mathbb{R}$ is a reward function defining rewards $r_t$ associated with transitions $(s_t, a_t, s_{t+1})$. $\gamma \in [0, 1)$ is a discount factor. Solving a MDP is equivalent to finding the optimal policy $\pi^*$ maximizing the expected return $J(\pi^*) = \mathbb{E}_{T, \pi^*}[\sum_{t=0}^{H} \gamma^t R(s_t, a_t, s_{t+1})]$ for some time horizon $H$, where actions are chosen according to $a_t = \pi^*(s_t)$. Lastly, let $\mathcal{S}$ be a Euclidean space, equipped with the standard Euclidean distance metric with an associated norm denoted by $\|\cdot\|$. The space of valid states is denoted by $\mathcal{F} \subseteq \mathcal{S}$.

## 3.2 EXPLORATION AS A PLANNING PROBLEM WITH RRT

The RRT algorithm (LaValle & Kuffner, 2001) provides a principled approach for planning in problems that cannot be solved directly (e.g. using inverse kinematics), but where it is possible to sample transitions. RRT builds a tree of *valid* transitions between states in $\mathcal{F}$, grown from a root $s_0$. As such, the tree $\mathbb{T}$ maintains information over *valid trajectories*. The exploration problem is defined by the pair $(\mathcal{F}, s_0)$. In RL environments with a known goal set $\mathcal{F}_{goal} \subseteq \mathcal{F}$ (e.g. MountainCar), the exploration problem is defined by $(\mathcal{F}, s_0, \mathcal{F}_{goal})$.

The RRT algorithm starts by sampling a random state $s_{rand} \in \mathcal{S}$, used to encourage exploration in a specific direction in the current iteration. This necessitates the first of two assumptions.

**Assumption 1.** *Random states can be sampled uniformly from the MDP state space $\mathcal{S}$.*

Sampled states are not required to be valid, thus sampling a random state is typically equivalent to trivially sampling a hyper-rectangle.

Then, the vertex $s_{near} = \arg\min_{s \in \mathbb{T}} \|s_{rand} - s_{near}\|$ is found. RRT attempts to expand the tree $\mathbb{T}$ from $s_{near}$ toward $s_{rand}$ by sampling an action $a \in \mathcal{A}$ according to a steering function $\Upsilon : \mathcal{S} \times \mathcal{S} \rightarrow \mathcal{A}$. In many planning scenarios, $\Upsilon$ samples randomly from $\mathcal{A}$. A forward step $s_{new} = f(s_{near}, a)$ is then simulated from $s_{near}$ using action $a$, where $f$ is defined by the transition dynamics. Being able to expand the tree from arbitrary $s_{near}$ relies on another assumption.

**Assumption 2.** *The environment state can be set to a previously visited state $s \in \mathbb{T}$.*

Although this assumption largely limits the algorithm to simulators, it has previously been used in Florensa et al. (2017); Nair et al. (2018); Ecoffet et al. (2019) for example; see discussion in Section 6 on overcoming the limitation to simulated environments.

$s_{new}$ is added as a new vertex of $\mathbb{T}$, alongside an edge $(s_{near}, s_{new})$ with edge information $a$. This process repeats until a sampling budget $k$ is exhausted or the goal set is reached (i.e. $s_{new} \in \mathcal{F}_{goal}$).

**Definition 1.** *A valid trajectory is a sequence $\tau = [s_0, a_0, s_1, \ldots, s_{t_\tau - 1}, a_{t_\tau - 1}, s_{t_\tau}]$ such that $(s_t, a_t, s_{t+1})$ is a valid transition and $s_t \in \mathcal{F}, \forall t \in \{0, 1, \ldots, t_\tau\}$. Whenever a goal set is defined, a successful valid trajectory end state satisfies $s_{t_\tau} \in \mathcal{F}_{goal}$.*

**Algorithm 1:** R3L exploration

**Input:** $s_0$, $k$: sampling budget
1   (optional) $\mathcal{F}_{goal}$, $p_g$: goal sampling prob.
**Output:** $\tau$: successful trajectory
2   Add root node $s_0$ to $\mathbb{T}$
3   **for** $i = 1 : k$ **do**
4   | $s_{rand} \leftarrow$ sample random state $s_{rand} \in \mathcal{S}$
5   | **if** $u \sim \mathbb{U}(0, 1) \leq p_g$ **then**
6   | | $s_{rand} \leftarrow$ sample $s_{rand}$ from $\mathcal{F}_{goal}$
7   | **end**
8   | $s_{near} \leftarrow$ find nearest node to $s_{rand}$ in $\mathbb{T}$
9   | $a \leftarrow$ sample $\pi_l(s_{near}, s_{rand} - s_{near})$
10  | $s_{new} \leftarrow$ execute $a$ in state $s_{near}$
11  | update $\pi_l$ with $(\{s_{near}, s_{new} - s_{near}\}, a)$
12  | Add node $s_{new}, a$ and edge $(s_{near}, s_{new})$ to $\mathbb{T}$
13  **end**
14  $\tau \leftarrow$ trajectory in $\mathbb{T}$ with max. cumulated reward

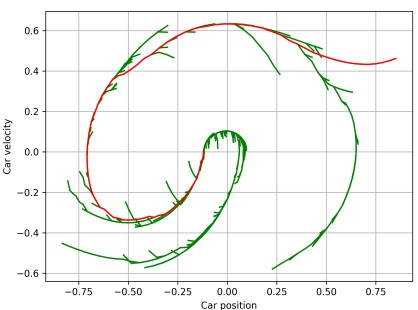

Figure 2: Example of R3L exploration on sparse MountainCar. Green segments are sampled transitions, executed in simulation. A successful solution found by R3L is displayed in red. State dimensions are normalized to $[-1, 1]$.

Once planning is finished, a successful valid trajectory can easily be generated from $\mathbb{T}$ by retrieving all nodes between the leaf $s_{leaf} \in \mathcal{F}_{goal}$ and the root. Because tree $\mathbb{T}$ is grown from valid transitions between states in $\mathcal{F}$, these trajectories are valid by construction.

### 3.3 R3L EXPLORATION: ADAPTING RRT TO RL

RRT is not directly applicable to RL problems. This subsection presents the necessary adaptations to the exploration phase of R3L, summed up in Algorithm 1. Figure 2 shows R3L's typical exploration behaviour.

**Local policy learning:** In classic planning problems, selecting actions extending $s_{near}$ towards $s_{rand}$ is easy, as these have a geometric interpretation or there is a known steering function. RL state spaces do not benefit from the same geometric properties, and properly selecting actions can be challenging. We solve this problem by defining a local policy $\pi_l$, which models the distribution of actions to transition from a state to a neighbouring goal state. Actions to extend a tree branch are sampled as $a \sim \pi_l(s_{near}, s_{rand} - s_{near})$. We formulate the problem of learning $\pi_l$ as supervised learning, where inputs are starting states $s_t$ augmented with the difference $s_{t+1} - s_t$, and targets are actions. The model is learned using transition data collected from previous tree expansion iterations. Results in this paper use Bayesian linear regression to represent $\pi_l$, but any supervised learning model can applied instead.

**Unknown dynamics:** RRT was designed for problems with known continuous dynamics $f$, but RL features unknown discrete transition dynamics. In R3L, $f$ is replaced with an environment interaction from $s_{near}$, with selected action $a$, resulting in a new state $s_{new}$. Since $(s_{near}, a, s_{new})$ is a real transition, it must be valid, and $s_{new}$ can be added to the tree $\mathbb{T}$.

**Biasing search with $\mathcal{F}_{goal}$:** Better exploration efficiency can be achieved if goal information is available. Indeed, the RRT framework allows for biasing exploration towards $\mathcal{F}_{goal}$, often resulting in faster exploration. This is achieved by sampling $s_{rand}$ from $\mathcal{F}_{goal}$ instead of $\mathcal{F}$ with low probability $p_g$, while the rest of the iteration remains unchanged.

Since R3L uses RRT to explore, the algorithm is most suitable for RL problems that are fully observable, exhibit sparse rewards and have continuous controls. R3L is applicable to other RL problems, but may not perform as well as methods tailored to specific cases.

### 3.4 POLICY INITIALIZATION FROM R3L DEMONSTRATIONS

Upon completion, R3L exploration yields a successful trajectory $\tau$, which may not be robust to various starting conditions and/or stochastic transitions. Converting successful trajectories into a policy is crucial to achieve robustness and enable further refinement with RL.

Policy initialization is applied to a set of successful trajectories $\boldsymbol{\tau} = \{\tau_i\}_1^N$ generated using $N$ runs of R3L exploration with different starting conditions. An imitation policy $\pi_0$ is learned by supervised

learning on transitions from $\tau$. Policy $\pi_0$ is then refined using traditional RL algorithms like TRPO. As shown in Figure 1, initializing policy parameters in the vicinity of a local optimum is crucial.

## 3.5 Exploration guarantees

The RL planning environment defines differential constraints of the form:

$$\dot{s} = f(s(t), a(t)), \quad s(t) \in \mathcal{F}, \quad a(t) \in \mathcal{A}. \tag{3}$$

Therefore, starting at $s_0$, the trajectory $\tau$ can be generated by forward integrating Eq. (3) with the applied actions. As with many RL problems, $a(t)$ is time-discretized resulting in a piecewise constant control function. This means $\tau$ is constructed of $n_\tau$ segments of fixed time duration $\Delta t$ such that the overall trajectory duration $t_\tau = n_\tau \cdot \Delta t$. Thus, $a(t)$ is defined as $a(t) = a_i \in \mathcal{A}$ where $t \in [(i-1) \cdot \Delta t, i \cdot \Delta t)$ and $1 \leq i \leq n_\tau$. Furthermore, as all transitions between states in $\tau$ are known, the trajectory return can be defined as $R_\tau = \sum_{t=0}^{n_\tau} \gamma^t R(s_t, a_t, s_{t+1})$.

R3L explores in state-action/trajectory space instead of policy parameter space. Furthermore, it is an effective exploration framework which provides *probabilistic completeness* (PC):

**Definition 2.** *A probabilistically complete planner finds a feasible solution (if one exists) with a probability approaching 1 in the limit of infinite samples.*

With the aforementioned dynamic characteristics, we prove that R3L exploration under the RL setting is PC. This is in stark contrast to the random walk exploration process, discussed in section 2, which is *not* PC. We begin with the following theorem, a modification of Theorem 2 from Kleinbort et al. (2019), which is applied to kinodynamic RRT where a goal set $\mathcal{F}_{goal}$ is defined.

**Theorem 1.** *Suppose that there exists a valid trajectory $\tau$ from $s_0$ to $\mathcal{F}_{goal}$ as defined in definition 1, with a corresponding piecewise constant control. The probability that R3L exploration fails to reach $\mathcal{F}_{goal}$ from $s_0$ after $k$ iterations is bounded by $ae^{-bk}$, for some constants $a, b > 0$.*

The proof, which is a modification of Theorem 2 from Kleinbort et al. (2019), can be found in Appendix S2. It should be noted that R3L exploration does not require an explicit definition for $\mathcal{F}_{goal}$ in order to explore the space. While in some path planning variants of RRT, $\mathcal{F}_{goal}$ is used to bias sampling, the main purpose of $\mathcal{F}_{goal}$ is to indicate that a solution has been found. Therefore, $\mathcal{F}_{goal}$ can be replaced by another implicit success criterion. In the RL setting, this can be replaced by a return-related criterion.

**Theorem 2.** *Suppose that there exists a trajectory with a return $R_\tau \geq \hat{R}, \hat{R} \in \mathbb{R}$. The probability that R3L exploration fails to find a valid trajectory from $s_0$ with $R_\tau \geq \hat{R}$ after $k$ iterations is bounded by $\hat{a}e^{-\hat{b}k}$, for some constants $\hat{a}, \hat{b} > 0$.*

*Proof.* The proof is straightforward. We augment each state in $\tau$ with the return to reach it from $s_0$:

$$s_n' = \begin{bmatrix} s_n \\ R_{s_n} \end{bmatrix}, \qquad \forall n = 1 : n_\tau, \tag{4}$$

where $R_{s_n=} = \sum_{t=0}^{n \leq n_\tau} \gamma^t R(s_t, a_t, s_{t+1})$. For consistency we modify the distance metric by simply adding a reward distance metric. With the above change in notation, we modify the goal set to $\mathcal{F}_{goal}^{RL} = \{(s, R_s) | s \in \mathcal{F}_{goal}, R_s \geq \hat{R}\}$, such that there is an explicit criterion for minimal return as a goal. Consequently, the exploration problem can be written for the augmented representation as $(\mathcal{F}, s_0^{RL}, \mathcal{F}_{goal}^{RL})$, where $s_0^{RL} = [s_0, 0]^\top$. Theorem 1 satisfies that R3L exploration can find a feasible solution to this problem within finite time, i.e. PC, and therefore the probability of not reaching $\mathcal{F}_{goal}^{RL}$ after $k$ iterations is upper-bounded by the exponential term $\hat{a}e^{-\hat{b}k}$, for some constants $\hat{a}, \hat{b} > 0$ ∎

We can now state our main result on the sampling complexity of the exploration process.

**Theorem 3.** *If trajectory exploration is probabilistically complete and satisfies an exponential convergence bound, the expected sampling complexity is finite and bounded such that*

$$\mathbb{E}[k] \leq \frac{\hat{a}}{4 \sinh^2 \frac{\hat{b}}{2}}, \tag{5}$$

*where $\hat{a}, \hat{b} > 0$.*

*Proof.* Theorem 2 provides an exponential bound for the probability the planner will fail in finding a feasible path. Hence, we can compute a bound for the expected number of iterations needed to find a solution, i.e. sampling complexity:

$$\mathbb{E}[k] \leq \sum_{k=1}^{\infty} k\hat{a}e^{-\hat{b}k} = \sum_{k=1}^{\infty} -\hat{a}\frac{de^{-\hat{b}k}}{d\hat{b}} = -\hat{a}\frac{d}{d\hat{b}}\sum_{k=1}^{\infty} e^{-\hat{b}k} = -\hat{a}\frac{d}{d\hat{b}}\frac{1}{e^{\hat{b}}-1} = \frac{\hat{a}}{4\sinh^2\frac{\hat{b}}{2}}, \quad (6)$$

where we used the relation $\sum_{k=1}^{\infty} e^{-\hat{b}k} = \frac{1}{e^{\hat{b}}-1}$. ∎

It is worth noting that while the sample complexity is bounded, the above result implies that its complexity varies according to problem-specific properties, which are encapsulated in the value of $\hat{a}$ and $\hat{b}$. Intuitively, $\hat{a}$ depends on the scale of the problem. It grows as $|\mathcal{F}_{goal}^{RL}|$ becomes smaller or as the length of the solution trajectory becomes longer. $\hat{b}$ depends on the probability of sampling states that will expand the tree in the right direction. It therefore shrinks as the dimensionality of $\mathcal{S}$ increases. We refer the reader to Appendix S2 for more details on the meaning of $\hat{a}, \hat{b}$ and the derivation of the tail bound in Theorem 1.

## 4 RELATED WORK

Exploration in RL has been extensively studied. Classic techniques typically rely on adding noise to actions (Mnih et al., 2015; Schulman et al., 2015) or to policy parameters (Plappert et al., 2018). However, these methods perform very poorly in settings with sparse rewards.

Intrinsic motivation tackles this problem by defining a new reward to direct exploration. Many intrinsic reward definitions were proposed, based on information theory (Oudeyer & Kaplan, 2008), state visit count (Lopes et al., 2012; Bellemare et al., 2016; Szita & Lőrincz, 2008; Fox et al., 2018), value function posterior variance (Osband et al., 2016a; Morere & Ramos, 2018), or model prediction error (Stadie et al., 2015; Pathak et al., 2017). Methods extending intrinsic motivation to continuous state and action spaces were recently proposed (Houthooft et al., 2016; Morere & Ramos, 2018). However, these approaches are less interpretable and offer no guarantees for the exploration of the state space.

Offering exploration guarantees, Bayesian optimization was adapted to RL in Wilson et al. (2014), to search over the space of policy parameters in problems with very few parameters. Recent work extends the method to functional policy representations (Vien et al., 2018), but results are still limited to toy problems and specific policy model classes.

Motion planning in robotics is predominantly addressed with sampling-based methods. This type of approach offers a variety of methodologies for exploration and solution space representation (e.g., *Probabilistic roadmaps* (PRM) (Kavraki et al., 1996), *Expansive space trees* (ESP) (Hsu et al., 1997) and *Rapidly-exploring random tree* (RRT) (Kuffner & LaValle, 2000)), which have shown excellent performance in path planning in high-dimensional spaces under dynamic constraints (LaValle & Kuffner, 2001; Hsu et al., 2002; Kavraki et al., 1996).

RL was previously combined with sampling-based planning to replace core elements of planning algorithms, such as PRM's point-to-point connection (Faust et al., 2018), local RRT steering function (Chiang et al., 2019) or RRT expansion policy (Chen et al., 2019). In contrast, the proposed method bridges the gap in the opposite direction, employing a sampling-based planner to generate demonstrations that kick-start RL algorithms and enhance their performance.

Accelerating RL by learning from demonstration was investigated in Niekum et al. (2015); Bojarski et al. (2016); Torabi et al. (2018). However, these techniques rely on user-generated demonstrations or a-priori knowledge of environment parameters. In contrast, R3L *automatically* generates demonstrations, with no need of an external expert.

## 5 EXPERIMENTS

In this section, we investigate (i) how learning a local policy $\pi_l$ and biasing search towards $\mathcal{F}_{goal}$ with probability $p_g$ affects R3L exploration, (ii) whether separating exploration from policy refinement

Table 1: Impact of learning local policy $\pi_l$ and biasing search towards $\mathcal{F}_{goal}$ with probability $p_g$ on R3L exploration. Results show the mean and standard deviation of successful trajectory length $|\tau|$ and number of timesteps required, computed over 20 runs.

| | | Goal bias ($p_g = 0.05$) | | Unbiased ($p_g = 0$) | |
| --- | --- | --- | --- | --- | --- |
| | | Learned $\pi_l$ | Random $\pi_l$ | Learned $\pi_l$ | Random $\pi_l$ |
| MountainCar | $|\tau|$ | $\mathbf{84.75 \pm 5.47}$ | $131.90 \pm 17.91$ | $86.75 \pm 11.82$ | $139.85 \pm 18.79$ |
| | timesteps | $\mathbf{895.65 \pm 190.70}$ | $4303.80 \pm 681.60$ | $928.90 \pm 204.0$ | $4447.55 \pm 417.10$ |
| Pendulum | $|\tau|$ | $73.10 \pm 12.86$ | $75.35 \pm 14.50$ | $\mathbf{67.05 \pm 15.30}$ | $77.90 \pm 12.62$ |
| | timesteps | $\mathbf{1108.65 \pm 155.29}$ | $2171.95 \pm 381.20$ | $1221.35 \pm 216.14$ | $2349.20 \pm 249.26$ |
| Acrobot | $|\tau|$ | $177.55 \pm 20.66$ | $\mathbf{163.95 \pm 19.19}$ | $173.5 \pm 24.22$ | $169.05 \pm 17.07$ |
| | timesteps | $15422.00 \pm 2624.16$ | $\mathbf{12675.20 \pm 2652.39}$ | $15792.65 \pm 3182.77$ | $13133.55 \pm 2060.51$ |
| Cartpole Swingup | $|\tau|$ | $\mathbf{217.35 \pm 53.09}$ | $319.20 \pm 58.78$ | $235.70 \pm 70.06$ | $348.35 \pm 80.09$ |
| | timesteps | $\mathbf{17502.75 \pm 13923.82}$ | $27186.70 \pm 12246.32$ | $23456.25 \pm 16792.11$ | $34482.20 \pm 12034.27$ |
| Reacher | $|\tau|$ | $32.70 \pm 13.55$ | $\mathbf{22.05 \pm 8.81}$ | $32.25 \pm 12.05$ | $25.30 \pm 11.02$ |
| | timesteps | $1445.80 \pm 1314.48$ | $\mathbf{838.85 \pm 846.94}$ | $1423.00 \pm 1030.07$ | $1034.55 \pm 1208.67$ |

is a viable and robust methodology in RL, (iii) whether R3L reduces the number of exploration samples needed to find good policies, compared with methods using classic and intrinsic exploration, and (iv) how R3L exploration can reduce the variance associated with policy gradient methods. All experiments make use of the Garage (Duan et al., 2016) and Gym (Brockman et al., 2016) frameworks. The experimental setup features the following tasks with sparse rewards: *Cartpole Swingup* ($\mathcal{S} \subseteq \mathbb{R}^4, \mathcal{A} \subseteq \mathbb{R}$), *MountainCar* ($\mathcal{S} \subseteq \mathbb{R}^2, \mathcal{A} \subseteq \mathbb{R}$), *Acrobot* ($\mathcal{S} \subseteq \mathbb{R}^4, \mathcal{A} \subseteq \mathbb{R}$), *Pendulum* ($\mathcal{S} \subseteq \mathbb{R}^2, \mathcal{A} \subseteq \mathbb{R}$), *Reacher* ($\mathcal{S} \subseteq \mathbb{R}^6, \mathcal{A} \subseteq \mathbb{R}^2$) *Fetch Reach* ($\mathcal{S} \subseteq \mathbb{R}^{13}, \mathcal{A} \subseteq \mathbb{R}^4$), and *Hand Reach* ($\mathcal{S} \subseteq \mathbb{R}^{78}, \mathcal{A} \subseteq \mathbb{R}^{20}$). The exact environment and reward definitions are described in Appendix S3.

**R3L exploration analysis** We first analyze the exploration performance of R3L in a limited set of RL environments, to determine the impact that learning policy $\pi_l$ has on exploration speed. We also investigate whether R3L exploration is viable in environments where no goal information is available. Table 1 shows the results of this analysis. Learning $\pi_l$ seems to greatly decrease the number of exploration timesteps needed on most environments. However, it significantly increases the number of timesteps on the acrobot and reacher environments. Results also suggest that learning $\pi_l$ helps R3L to find shorter trajectories on the same environments, which is a desirable property in many RL problems. Biasing R3L exploration towards the goal set $\mathcal{F}_{goal}$ helps finding successful trajectories faster, as well as reducing their length. However, R3L exploration without goal bias is still viable in all cases. Although goal information is not given in the classic MDP framework, it is often available in real-world problems and can be easily utilized by R3L. Lastly, successful trajectory lengths have low variance, which suggests R3L finds consistent solutions.

**Comparison to classic and intrinsic exploration on RL benchmarks** We examine the rates at which R3L learns to solve several RL benchmarks, and compare them with state-of-the-art RL algorithms. Performance is measured in terms of undiscounted returns and aggregated over 10 random seeds, sampled at random for each environment. We focus on domains with sparse rewards, which are notoriously difficult to explore for traditional RL methods. Our experiments focus on the widely-used methods TRPO (Schulman et al., 2015) and DDPG (Lillicrap et al., 2015). R3L-TRPO and R3L-DDPG are compared to the baseline algorithms with Gaussian action noise. As an additional baseline we include VIME-TRPO (Houthooft et al., 2016). VIME is an exploration strategy based on maximizing information gain about the agent's belief of the environment dynamics. It is included to show that R3L can improve on state-of-the-art exploration methods as well as naive ones, even though the return surface for VIME-TRPO is no longer flat, unlike Figure 1. The exact experimental setup is described in Appendix S3.2. The R3L exploration phase is first run to generate training trajectories for all environments. The number of environment interactions during this phase is accounted for in the results, displayed as an offset with a vertical dashed black line. The average performance achieved by these trajectories is also reported as a guideline, with the exception of *Cartpole Swingup* where doing so does not make sense. RL is then used to refine a policy pretrained with these trajectories.

Figure 3 shows the median and interquartile range for all methods. R3L is very competitive with both vanilla and VIME baselines. It converges faster and achieves higher performance at the end of the experiment. In most cases, the upper quartile for our method begins well above the minimum return, indicating that R3L exploration and pre-training are able to produce successful though not optimal policies. R3L-DDPG performance, for the majority of problems, starts significantly above the minimum return, plunges due to the inherent instability of DDPG, but eventually recovers, indicating that R3L pre-training can help mitigate the instability. It is worth noting that R3L's lower quartile is considerably higher than that of baselines. Indeed, for many of the baselines the lower quartile

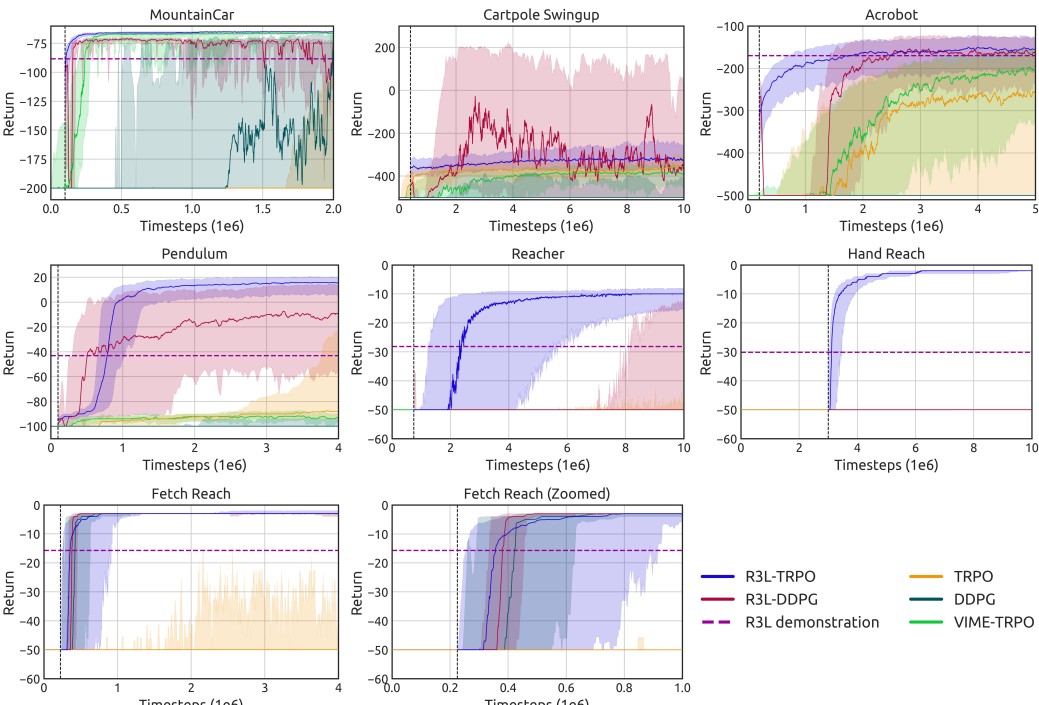

Figure 3: Results for classic control tasks, comparing our proposed method (R3L-TRPO/DDPG), vanilla TRPO/DDPG, and VIME-TRPO. Trendlines are the medians and shaded areas are the interquartile range, taken over 10 randomly chosen seeds. Also shown is the average undiscounted return of successful trajectories generated with R3L exploration. The dashed offset at the start of R3L-TRPO/DDPG reflects the number of timesteps spent on R3L exploration.

takes a long time to improve on the minimum return, and in some cases it never manages to do so at all. This is a common problem in sparse reward RL, where there is no alternative but to search the parameter space randomly until the first successful trajectory is found, as explained in Section 2. While a few random seeds will by chance find a successful trajectory quickly (represented by the quickly rising upper quartile), others take a long time (represented by the much slower rise of the median and lower quartile). In other words, R3L-TRPO/DDPG is much more robust to random policy initialization and to the random seed than standard RL methods. This is because R3L is able to use automatically generated demonstrations to initialize RL policy parameters to a region with informative return gradients.

# 6 CONCLUSION

We proposed Rapidly Randomly-exploring Reinforcement Learning (R3L), an exploration paradigm for leveraging planing algorithms to automatically generate successful demonstrations, which can be converted to policies then refined by classic RL methods. We provided theoretical guarantees of R3L finding solutions, as well as sampling complexity bounds. Empirical results show that R3L outperforms classic and intrinsic exploration techniques, requiring only a fraction of exploration samples and achieving better asymptotic performance.

As future work, R3L could be extended to real-world problems by leveraging recent advances on bridging the gap between simulation and reality (Peng et al., 2018). Respecting Assumption 2, a policy would first be trained on a simulator and then transferred to the real-world. Exploration in high-dimensional tasks is also challenging as stated in Theorem 3 and confirmed experimentally by increased R3L exploration timesteps. Exploiting external prior knowledge and/or the structure of the problem can benefit exploration in high-dimensional tasks, and help make R3L practical for problems such as Atari games. Lastly, recent advances in RRT (Chiang et al., 2019) and learning from demonstration (Torabi et al., 2018) could also improve R3L.

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

# Reinforcement Learning with Probabilistically Complete Exploration: Supplementary Material

## S1 APPENDIX A: RRT ALGORITHM PSUEDO-CODE

In this section, we provide pseudo-code of the classic RRT algorithm.

---

**Algorithm S1:** Rapidly-exploring Random Trees (RRT)

**Input:** $s_{init}$
1       $k$: sampling budget
2       $\delta t$: Euler integration time interval
**Output:** $\mathbb{T}$
3  $\mathbb{T}.\text{init}(s_{init})$
4  **for** $i = 1 : k$ **do**
5     $s_{rand} \leftarrow \text{RANDOM\_UNIFORM}(S)$
6     **if** $s_{rand} \notin \mathcal{F}$ **then**
7       | pass
8     **end**
9     $s_{near} \leftarrow \arg\min_{s \in \mathbb{T}} \|s_{rand} - s_{near}\|$       /* Find nearest vertex */
10    $a \leftarrow \Upsilon(s_{near}, s_{rand})$            /* Sample action */
11    $s_{new} \leftarrow s_{near} + \Delta t \cdot f(s_{near}, a)$    /* Propagate to new state, Eq. (3) */
12    **if** *VALID\_TRANSITION*$(s_{near}, s_{new})$ **then**
13       $\mathbb{T}.\text{add\_vertex}(s_{new})$
14       $\mathbb{T}.\text{add\_edge}(s_{near}, s_{new})$
15    **end**
16  **end**

---

## S2 APPENDIX B: PROOF OF THEOREM 1

This appendix proves Theorem 1 which shows that planning using RRT under differential constraints is probabilistically complete. The following proof is a modification of Theorem 2 from Kleinbort et al. (2019), where completeness of RRT in the RL setting is maintained without the need to explicitly sample a duration for every action.

Equation 3 defines the environment's differential constraints. In practice, Eq. (3) is approximated by an Euler integration step. With the interval $[0, t_\tau]$ divided into $l >> n_\tau$ equal time intervals of duration $h$ with $t_\tau = l \cdot h$. Eq. (3) is then approximated by an Euler integration step, where the transition between consecutive time steps is given by:

$$
\begin{aligned}
s_{n+1} &= s_n + f(s_n, a_n) \cdot h, \qquad s_n, s_{n+1} \in \tau, \\
s.t. \lim_{l \to \infty, h \to 0} &\|s_n - \tau(n \cdot h)\| = 0.
\end{aligned}
\tag{S1}
$$

Furthermore, we define $\mathcal{B}_r(s)$ as a ball with a radius $r$ centered at $s$ for any given state $s \in \mathcal{S}$.

We assume that the planning environment is Lipschitz continuous in both state and action, constraining the rate of change of Eq. (S1). Formally, there exists two positive constants $K_s, K_a > 0$, such that $\forall s_0, s_1 \in \mathcal{F}, a_0, a_1 \in \mathcal{A}$ :

$$\|f(s_0, a_0) - f(s_0, a_1)\| \le K_a \|a_0 - a_1\|, \tag{S2}$$

$$\|f(s_0, a_0) - f(s_1, a_0)\| \le K_s \|s_0 - s_1\|. \tag{S3}$$

**Lemma 1.** *For two trajectories $\tau, \tau'$, where $s_0 = \tau(0)$ and $s_0' = \tau'(0)$ such that $\|s_0 - s_0'\| \le \delta_s$ with $\delta_s$ a positive constant. Suppose that for each trajectory a piecewise constant action is applied, so that $\Upsilon(t) = a$ and $\Upsilon'(t) = a'$ is fixed during a time period $T \ge 0$. Then $\|\tau(T) - \tau'(T)\| \le e^{K_s T} \delta_s + T K_a e^{K_s T} \|a - a'\|$.*

The proof for Lemma 1 is given in Lemma 2 of (Kleinbort et al., 2019). Intuitively, this bound is derived from compounding worst-case divergence between $\tau$ and $\tau'$ at every Euler step along $T$ which leads to an overall exponential dependence.

Using Lemma 1, we want to provide a lower bound on the probability of choosing an action that will expand the tree successfully. We note that this scenario assumes that actions are drawn uniformly from $\mathcal{A}$, i.e. there is no steering function [2]. When better estimations of the steering function are available, as described in 3.3, the performance of RRT significantly improves.

**Definition 3.** *A trajectory $\tau$ is defined as $\delta$-clear if for $\delta_{clear} > 0$, $\mathcal{B}_{\delta_{clear}}(\tau(t)) \in \mathcal{F}$ for all $t \in [0, t_\tau]$.*

**Lemma 2.** *Suppose that $\tau$ is a valid trajectory from $\tau(0) = s_0$ to $\tau(t_\tau) = s_{goal}$ with a duration of $t_\tau$ and a clearance of $\delta$. Without loss of generality, we assume that actions are fixed for all $t \in [0, t_\tau]$, such that $\Upsilon(t) = a \in \mathcal{A}$.*

*Suppose that RRT expands the tree from a state $s'_0 \in \mathcal{B}_{(\kappa\delta - \epsilon)}(s_0)$ to a state $s'_{goal}$, for any $\kappa \in (0, 1]$ and $\epsilon \in (0, \kappa\delta)$ we can define the following bound:*

$$\Pr[s'_{goal} \in \mathcal{B}_{\kappa\delta}(s_{goal})] \geq \frac{\zeta_{|\mathcal{S}|} \cdot \frac{\kappa\delta - e^{K_s t_\tau}(\kappa\delta - \epsilon)}{K_a t_\tau e^{K_s t_\tau}}}{|\mathcal{A}|}.$$

*Here, $\zeta_{|\mathcal{S}|} = |\mathcal{B}_1(\cdot)|$ is the Lebesgue measure for a unit circle in $\mathcal{S}$.*

*Proof.* We denote $\tau'$ a trajectory that starts from $s'_0$ and is expanded with an applied random action $a_{rand}$. According to Lemma 1,

$$\|\tau(t) - \tau'(t)\| \leq e^{K_s t}\delta_s + t K_a e^{K_s t}\|a - a_{rand}\| \leq e^{K_s t}(\kappa\delta - \epsilon) + t K_a e^{K_s t}\|a - a_{rand}\|, \qquad \forall t \in [0, t_\tau],$$

where $\delta_s \leq \kappa\delta - \epsilon$ since $s'_0 \in \mathcal{B}_{\kappa\delta - \epsilon}(s_0)$. Now, we want to find $\|a - a_{rand}\|$ such that the distance between the goal points of these trajectories, i.e. in the worst-case scenario, is bounded:

$$e^{K_s t_\tau}(\kappa\delta - \epsilon) + t_\tau K_a e^{K_s t_\tau}\|a - a_{rand}\| < \kappa\delta.$$

After rearranging this formula, we can obtain a bound for $\|a - a_{rand}\|$:

$$\Delta a = \|a - a_{rand}\| < \frac{\kappa\delta - e^{K_s t_\tau}(\kappa\delta - \epsilon)}{t_\tau K_a e^{K_s t_\tau}}.$$

Assuming that $a_{rand}$ is drawn out of a uniform distribution, the probability of choosing the proper action is

$$p_a = \frac{\zeta_{|\mathcal{S}|} \cdot \frac{\kappa\delta - e^{K_s t_\tau}(\kappa\delta - \epsilon)}{t_\tau K_a e^{K_s t_\tau}}}{|\mathcal{A}|}, \tag{S4}$$

where $\zeta_{|\mathcal{S}|}$ is used to account for the degeneracy in action selection due to the dimensionality of $\mathcal{S}$. We note that $\epsilon \in (0, \kappa\delta)$ guarantees $p_a \geq 0$, thus a valid probability> ∎

Equation S4 provides a lower bound for the probability of choosing the suitable action. The following lemma provides a bound on the probability of randomly drawing a state that will expand the tree toward the goal.

**Lemma 3.** *Let $s \in \mathcal{S}$ be a state with clearance $\delta$, i.e. $\mathcal{B}_\delta(s) \in \mathcal{F}$. Suppose that for an RRT tree $\mathbb{T}$ there exist a vertex $v \in \mathbb{T}$ such that $v \in \mathcal{B}_{2\delta/5}(s)$. Following the definition in Section 3.2, we denote $s_{near} \in \mathbb{T}$ as the closest vertex to $s_{rand}$. Then, the probability that $s_{near} \in \mathcal{B}_\delta(s)$ is at least $|\mathcal{B}_{\delta/5}(s)|/|S|$.*

*Proof.* Let $s_{rand} \in \mathcal{B}_{\delta/5}(s)$. Therefore the distance between $s_{rand}$ and $v$ is upper-bounded by $\|s_{rand} - v\| \leq 3\delta/5$. If there exists a vertex $s_{near}$ such that $s_{near} \neq v$ and $\|s_{rand} - s_{near}\| \leq \|s_{rand} - v\|$, then $s_{near} \in \mathcal{B}_{3\delta/5}(s_{rand}) \subset \mathcal{B}_\delta(s)$. Hence, by choosing $s_{rand} \in \mathcal{B}_{\delta/5}(s)$, we are guaranteed $s_{near} \in \mathcal{B}_\delta(s)$. As $s_{rand}$ is drawn uniformly, the probability for $s_{rand} \in \mathcal{B}_{\delta/5}(s)$ is $|\mathcal{B}_{\delta/5}(s)|/|S|$. ∎

---

[2]The function $steer : \mathcal{S} \times \mathcal{S} \to \mathcal{A}$ returns an action $a_{steer}$ given two states $srand$ and $s_{near}$ such that $a_{steer} = \arg\min_{a \in \mathcal{A}}\|s_{rand} - (s_{near} + \Delta t \cdot f(s_{near}, a))\|$ s.t. $\|\Delta t \cdot f(s_{near}, a)\| < \eta$, for a prespecified $\eta > 0$ (Karaman & Frazzoli, 2011).

We can now prove the main theorem.

**Theorem 1.** *Suppose that there exists a valid trajectory $\tau$ from $s_0$ to $\mathcal{F}_{goal}$ as defined in definition 1, with a corresponding piecewise constant control. The probability that RRT fails to reach $\mathcal{F}_{goal}$ from $s_0$ after $k$ iterations is bounded by $ae^{-bk}$, for some constants $a, b > 0$.*

*Proof.* Lemma 2 puts bound on the probability to find actions that expand the tree from one state to another in a given time. As we assume that a valid trajectory exists, we can assume that the probability defined in Lemma 2 is non-zero, i.e. $p_a > 0$, hence:

$$\kappa\delta - e^{K_s \Delta t}(\kappa\delta - \epsilon) > 0, \tag{S5}$$

where we set $\kappa = 2/5$ and $\epsilon = 5^{-2}$ as was also done in (Kleinbort et al., 2019). We additionally require that $\Delta t$, which is typically defined as an RL environment parameter, is chosen accordingly so to ensure that Eq. (S5) holds, i.e. $K_s \Delta t < \log\left(\frac{\kappa\delta}{\kappa\delta - \epsilon}\right)$.

We cover $\tau$ with balls of radius $\delta = \min\{\delta_{goal}, \delta_{clear}\}$, where $\mathcal{B}_{\delta_{goal}} \subseteq \mathcal{F}_{goal}$. The balls are spaced equally in time with the center of the $i^{th}$ ball is in $c_i = \tau(\Delta t \cdot i), \forall i = 0 : m$, where $m = t_\tau / \Delta t$. Therefore, $c_0 = s_0$ and $c_m = s_{goal}$. We now examine the probability of RRT propagating along $\tau$. Suppose that there exists a vertex $v \in \mathcal{B}_{2\delta/5}(c_i)$, we need to bound the probability $p$ that by taking a random sample $s_{rand}$, there will be a vertex $s_{near} \in \mathcal{B}_\delta(c_i)$ such that $s_{new} \in \mathcal{B}_{2\delta/5}(c_{i+1})$. Lemma 3 provides a lower bound for the probability that $s_{near} \in \mathcal{B}_\delta(c_i)$, given that there exists a vertex $v \in \mathcal{B}_{2\delta/5}(c_i)$, of $|\mathcal{B}_{\delta/5}(s)|/|S|$. Lemma 2 provide a lower bound for the probability of choosing an an action from $s_{near}$ to $s_{new}$ of $\rho \equiv \frac{\zeta_{|S|} \cdot \frac{\kappa\delta - e^{K_s \Delta t}(\kappa\delta - \epsilon)}{\Delta t K_a e^{K_s \Delta t}}}{|\mathcal{A}|} > 0$, where we have substituted $t_\tau$ with $\Delta t$. Consequently, $p \geq (|\mathcal{B}_{\delta/5}(s)| \cdot \rho)/|S|$.

For RRT to recover $\tau$, the transition between consecutive circles must be repeated $m$ times. This stochastic process can be described as a binomial distribution, where we perform $k$ trials (randomly choosing $s_{rand}$), with $m$ successes (transition between circles) and a transition success probability $p$. The probability mass function of a binomial distribution is $\Pr(X = m) = \Pr(m; k, p) = \binom{k}{m}p^m(1-p)^{k-m}$. We use the cumulative distribution function (CDF) to represent the upper bound for failure, i.e. the process was unable to perform $m$ steps, which can be expressed as:

$$\Pr(X < m) = \sum_{i=0}^{m-1} \binom{k}{i} p^i (1-p)^{k-i}. \tag{S6}$$

Using Chernoff's inequality we derive the tail bounds of the CDF when $m \leq p \cdot k$:

$$\Pr(X < m) \leq \exp\left(-\frac{1}{2p}\frac{(kp-m)^2}{k}\right) \tag{S7}$$

$$= \exp\left(-\frac{1}{2}kp + m - \frac{m^2}{kp}\right) \tag{S8}$$

$$\leq e^m e^{-\frac{1}{2}pk} = ae^{-bk}. \tag{S9}$$

In the other case, where $p < m/k < 1$, the upper bound is given by (Arratia & Gordon, 1989):

$$\Pr(X < m) \leq \exp\left(-k\mathcal{D}\left(\frac{m}{k} \parallel p\right)\right), \tag{S10}$$

where $\mathcal{D}$ is the relative entropy such that

$$D\left(\frac{m}{k} \parallel p\right) = \frac{m}{k}\log\frac{\frac{m}{k}}{p} + (1 - \frac{m}{k})\log\frac{1 - \frac{m}{k}}{1 - p}.$$

Rearranging $\mathcal{D}$, we can rewrite S10 as follows:

$$\Pr(X < m) \leq \exp\left(-k\left(\frac{m}{k}\log\left(\frac{m}{kp}\right) + \frac{k-m}{k}\log\left(\frac{1-\frac{m}{k}}{1-p}\right)\right)\right) \tag{S11}$$

$$= \exp\left(-m\log\left(\frac{m}{kp}\right)\right)\exp\left(-k\log\left(\frac{1-\frac{m}{k}}{1-p}\right)\right)\exp\left(m\log\left(\frac{1-\frac{m}{k}}{1-p}\right)\right) \tag{S12}$$

$$= \exp\left(-m\log\left(\frac{m(1-p)}{kp(1-\frac{m}{k})}\right)\right)\exp\left(-k\log\left(\frac{1-\frac{m}{k}}{1-p}\right)\right) \tag{S13}$$

$$\leq \exp\left(-k\log\left(\frac{1-\frac{m}{k}}{1-p}\right)\right) \tag{S14}$$

$$\leq \exp\left(-k\log\left(\frac{0.5}{1-p}\right)\right) \tag{S15}$$

$$\leq e^{-kp} = ae^{-bk}, \tag{S16}$$

where (S14) is justified for worst-case scenario where $p = m/k$, (S15) uses the fact that $p < m/k < 0.5$, hence $1 - m/k > 0.5$. The last step, (S16) is derived from the first term of the Taylor expansion of $\log\left(\frac{1}{1-p}\right) = \sum_{j=1}^{\infty}\frac{p^j}{j}$.

As $p$ and $m$ are fixed and independent of $k$, we show that the expression for $\Pr(X < m)$ decays to zero exponentially with $k$, therefore RRT is probabilistically complete. ∎

It worth noting that as expected the failure probability $\Pr(X < m)$ depends on problem-specific properties, which give rise to the values of $a$ and $b$. Intuitively, $a$ depends on the scale of the problem such as volume of the goal set $|\mathcal{F}_{goal}^{RL}|$ and how complex and long the solution needs to be, as evident in Eq. (S9). More importantly, $b$ depends on the probability $p$. Therefore, it is a function of the dimensionality of $\mathcal{S}$ (through the probability of sampling $s_{rand}$) and other environment parameters such as clearance (defined by $\delta$) and dynamics (via $K_s$, $K_a$), as specified in Eq. (S4).

## S3 APPENDIX C: EXPERIMENTAL SETUP

All experiments were run using a single 2.2GHz core and a GeForce GTX 1080 Ti GPU.

### S3.1 ENVIRONMENTS

All environments are made available in supplementary code. Environments are based on Gym (Brockman et al., 2016), with modified sparse reward functions and state spaces. All environments emit a $-1$ reward per timestep unless noted otherwise. The environments have been further changed from Gym as follows:

- *Cartpole Swingup*- The state space $\mathcal{S} \subseteq \mathbb{R}^4$ consists of states $s = \left[x, \theta, \dot{x}, \dot{\theta}\right]$ where $x$ is cart position, $\dot{x}$ is cart linear velocity, $\theta$ is pole angle (measuring from the $y$-axis) and $\dot{\theta}$ pole angular velocity. Actions $\mathcal{A} \subseteq \mathbb{R}$ are force applied on the cart along the $x$-axis. The goal space $\mathcal{F}_{goal}$ is $\{s \in \mathcal{S} \mid \cos\theta > 0.9\}$. Note that reaching the goal space does not terminate an episode, but yields a reward of $\cos\theta$. Time horizon is $H = 500$. Reaching the bounds of the rail does not cause failure but arrests the linear movement of the cart.

- *MountainCar*- The state space $\mathcal{S} \subseteq \mathbb{R}^2$ consists of states $s = [x, \theta]$ where $x$ is car position and $\dot{x}$ is car velocity. Actions $\mathcal{A} \subseteq \mathbb{R}$ are force applied by the car engine. The goal space $\mathcal{F}_{goal}$ is $\{s \in \mathcal{S} \mid x \geq 0.45\}$. Time horizon is $H = 200$.

- *Acrobot*- The state space $\mathcal{S} \subseteq \mathbb{R}^4$ consists of states $s = \left[\theta_0, \theta_1, \dot{\theta}_0, \dot{\theta}_1\right]$ where $\theta_0, \theta_1$ are the angles of the joints (measuring from the $y$-axis and from the vector parallel to the $1^{st}$ link, respectively) and $\dot{\theta}_0, \dot{\theta}_1$ are their angular velocities. Actions $\mathcal{A} \subseteq \mathbb{R}$ are torque applied on the $2^{nd}$ joint. The goal space $\mathcal{F}_{goal}$ is $\{s \in \mathcal{S} \mid -\cos\theta_0 - \cos(\theta_0 + \theta_1) > 1.9\}$. In other words, the set of states where the end of the second link is at a height $y > 1.9$. Time horizon is $H = 500$.

- *Pendulum*- The state space $\mathcal{S} \subseteq \mathbb{R}^2$ consists of states $s = \left[\theta, \dot{\theta}\right]$ where $\theta$ is the joint angle (measured from the $y$-axis) and $\dot{\theta}$ is the joint angular velocity. Actions $\mathcal{A} \subseteq \mathbb{R}$ are torque applied on the joint. The goal space $\mathcal{F}_{goal}$ is $\{s \in \mathcal{S} \mid \cos\theta > 0.99\}$. Note that reaching the goal space does not terminate an episode, but yields a reward of $\cos\theta$. Time horizon is $H = 100$.

- *Reacher*- The state space $\mathcal{S} \subseteq \mathbb{R}^6$ consists of states $s = \left[\theta_0, \theta_1, x, y\dot{\theta}_0, \dot{\theta}_1\right]$ where $\theta_0, \theta_1$ are the angles of the joints, $(x, y)$ are the coordinates of the target and $\dot{\theta}_0, \dot{\theta}_1$ are the joint angular velocities. Actions $\mathcal{A} \subseteq \mathbb{R}^2$ are torques applied at the 2 joints. The goal space $\mathcal{F}_{goal}$ is the set of states where the end-effector is within a distance of $0.01$ from the target. Time horizon is $H = 50$.

- *Fetch Reach*- A high-dimensional robotic task where the state space $\mathcal{S} \subseteq \mathbb{R}^{13}$ consists of states $s = [gripper\_pos, finger\_pos, gripper\_state, finger\_state, goal\_pos]$ where $gripper\_pos, gripper\_vel$ are the Cartesian coordinates and velocities of the Fetch robot's gripper, $finger\_state, finger\_vel$ are the two-dimensional position and velocity of the gripper fingers, and $goal\_pos$ are the Cartesian coordinates of the goal. Actions $\mathcal{A} \subseteq \mathbb{R}^4$ are relative target positions of the gripper and fingers, which the MuJoCo controller will try to achieve. The goal space $\mathcal{F}_{goal}$ is the set of states where the end-effector is within a distance of $0.05$ from $goal\_pos$. Time horizon is $H = 50$. Note that this problem is harder than the original version in OpenAI Gym, as we only sample $goal\_pos$ that are far from the gripper's initial position.

- *Hand Reach*- A high-dimensional robotic task where the state space $\mathcal{S} \subseteq \mathbb{R}^{78}$ consists of states $s = [joint\_pos, joint\_vel, fingertip\_pos, goal\_pos]$ where $joint\_pos, joint\_vel$ are the angles and angular velocities of the Shadow hand's 24 joints, $fingertip\_pos$ are the Cartesian coordinates of the 5 fingertips, and $goal\_pos$ are the Cartesian coordinates of the goal positions for each fingertip. Actions $\mathcal{A} \subseteq \mathbb{R}^{20}$ are absolute target angles of the 20 controllable joints, which the MuJoCo controller will try to achieve. The goal space $\mathcal{F}_{goal}$ is the set of states where all fingertips are simultaneously within a distance of $0.02$ from their respective goals. Time horizon is $H = 50$.

## S3.2 EXPERIMENTAL SETUP AND HYPER-PARAMETER CHOICES

All experiments feature a policy with 2 fully-connected hidden layers of 32 units each with tanh activation, with the exception of *Reacher*, for which a policy network of 4 fully-connected hidden layers of 128 units each with relu activation is used. For all environments we use a linear feature baseline for TRPO.

Default values are used for most hyperparameters. A discount factor of $\gamma = 0.99$ is used in all environments. For VIME, hyperparameters values reported in the original paper are used, and the implementation published by the authors was used.

For TRPO, default hyperparameter values and implementation from Garage are used: KL divergence constraint $\delta = 10^{-2}$, and Gaussian action noise $\mathcal{N}(0, 0.3^2)$.

In comparisons with VIME-TRPO and vanilla TRPO, the R3L goal sampling probability $p_g$ is set to $0.05$, as proposed in Urmson & Simmons (2003). Goal sets $\mathcal{F}_{goal}$ are defined in Appendix S3.1 for each environment. In all experiments, the local policy $\pi_l$ learned during R3L exploration uses Bayesian linear regression with prior precision $0.1$ and noise precision $1.0$, as well as 300 random Fourier features (Rahimi & Recht, 2008) approximating a square exponential kernel with lengthscale $0.3$.

