# OpenReview forum: "Reinforcement Learning with Probabilistically Complete Exploration"
_ICLR.cc/2020/Conference — Reject_

### Official Review · AnonReviewer2 · 2019-10-17
**Official Blind Review #2**

**Rating:** 3

**Review:**

Reinforcement Learning with Probabilistically Complete Exploration
==========================================================

This paper proposes an exploration technique for planning from simulator.
Roughly speaking, the algorithm uses some initial budget to sample random states and generate some effective demonstration trajectory.
Once this trajectory is found, it can be used to form an initialization for a policy gradient method.
This leads to improved performance on some mujoco tasks.


There are several things to like about this paper:
- The problem of efficient exploration in large-scale RL is a big outstanding hole. Particularly finding methods that are compatible with state of the art policy gradient approaches.
- The proposed algorithm is sensible, and seems to use a reasonable heuristic from planning to generate good "kickstart" for policy gradient methods.
- The general quality of the writing and presentation is pretty good.
- It's great to see code released.

However, there are some places where this paper falls down:
- Assumptions 1 (and particularly 2) are *not* part of the standard RL problem... but actually show that this is a proposal for planning given a simulator. This is a different problem setting and the distinction is really not clear from the first (several) pages. Further, although the assumptions are stated clearly, I think that this leads to some unfair comparisons (even if the sampled states are taken from the X-axis budget in plots).
  - Note that this paper is far from the only one that is a bit sloppy on this distinction... and, of course, you can still use an RL *algorithm* to solve the planning *problem*... but it's not clear you can use a planning algorithm to solve the RL problem... and that's what this paper claims... but then Assumption 2 essentially just reduces the RL problem to the planning problem!
- The claims of "probabilistic completeness" are not particularly insightful, in fact, the same is also true of Q-learning with epsilon-greedy dithering! The point of efficient exploration would be that you find this stuff quickly... and I'm not really convinced that this method always would. The quality of \hat{a}, \hat{b} seems like it should be very important... but I don't get much insight to that spelled out in the paper.
- The computational evaluations are not particularly insightful, in that they seems to not give much insight into exactly what is happening. I also wonder whether they are really "fair" comparisons given the planning vs RL distinction.


Overall, I think that there are interesting pieces to the paper, and the underlying algorithm is also interesting.
For me, the confusion between the planning and RL setting is impossible to move past... particularly since the exploration challenges can be distinct in these domains.
For this reason, I don't think the paper is ready for publication.


**Experience Assessment:**

I have published in this field for several years.

**Review Assessment: Checking Correctness Of Derivations And Theory:**

I assessed the sensibility of the derivations and theory.

**Review Assessment: Checking Correctness Of Experiments:**

I carefully checked the experiments.

**Review Assessment: Thoroughness In Paper Reading:**

I read the paper at least twice and used my best judgement in assessing the paper.

---

> ### Author Response · Authors · 2019-11-11
> **Response to Official Blind Review #2**
>
> We would like to thank Reviewer 2 for the thorough review and helpful suggestions.
>
> We understand Reviewer 2's concerns that there is some confusion in the paper between the planning and RL settings. This is perhaps due to lack of clarity in the writing. The paper deals with two separate but related problems:
>   * The original RL problem. The solution to this problem is a policy that achieves high expected return.
>   * A related planning problem, which can be constructed based on the RL problem. The solution to this problem is a valid trajectory that connects a single initial state with a success state. Assumptions 1 and 2 only apply to this problem.
>
> The planning algorithm is used only to solve the related planning problem (2), and its solutions (trajectories) are used to initialise a policy. The original RL problem (1) is retained, and is solved with a typical RL algorithm, not with a planning algorithm. The construction and solution of the planning problem (2) serve only to generate demonstrations (without resorting to a human or heuristic expert), which the learning-from-demonstration setting usually takes for granted. The idea is that we can use the solution of a planning problem to initialise a solver for the more difficult RL problem.
>
> Regarding probabilistic completeness, the important point is that policy gradient RL is *not* probabilistically complete in the sparse reward setting (cf. Equation 2), and given a bad initialisation it may not find a solution even in the limit of infinite iterations. In general, no method can guarantee a quick solution to an arbitrarily complex MDP, but policy gradient can't guarantee even a slow solution.
>
> Regarding $\hat{a}$ and $\hat{b}$, we have updated the paper to clarify what properties of the MDP they reflect.

---

> > ### Comment · AnonReviewer2 · 2019-11-14
> > **Thanks**
> >
> > Thank you for your response.
> >
> > It will take me some time to review the updated paper.
> > I still have concerns over the treatment of (1) and (2), with respect to the assumptions that are now required.
> >
> > But I will have a look through the new material and see if it causes me to change to accept.
> >
> > Many thanks

---

### Official Review · AnonReviewer1 · 2019-10-21
**Official Blind Review #1**

**Rating:** 6

**Review:**

This paper proposes merging the 'Rapidly-exploring Random Tree' algorithm and RL to inform exploration. I think this is an interesting and novel idea, and the paper is quite well written, although some issues remain.

Firstly, is this algorithm applicable to all RL problems? The assumptions seem quite strong, and furthermore it assumes the existence of a 'goal' state, which is not always the case. I would have liked to see the performance on more challenging domains, like the atari suite, to be fully convinced. I think the question of where and when this algorithm is appropriate needs to be significantly expanded upon.

Furthermore it seems that in a more challenging domain the memory requirements would explode, is that correct? In some sense it seems similar to this paper: https://arxiv.org/abs/1606.04460 which should be discussed.

The name 'Rapidly Randomly-exploring Reinforcement Learning' is a bit jarring since there are no theoretical guarantees about the regret of this approach to indicate that this actually is 'rapid' exploration, at least in the sense that people use it in RL (eg, being efficient with respect to data in the exploration-exploitation tradeoff).

In algorithm 1 there are several confusing aspects:

* I don't understand what is meant by "project srand onto Fg", what is the projection and how is it performed?
* How do you "find nearest node to srand in T"? How is nearness measured? Note that in RL two similar states (eg l2 distance) may be very far apart by many metrics.
* What is "πl(snear, srand − snear)"? How is it parameterized? I see the discussion in the appendix but I think it needs more discussion up front.
* "update π", update how?

Do the theoretical results rely on non trivial measure for the goal states and / or compactness? I'm thinking of an example of a single point (ie measure zero) being the goal state and RRT trying to search for it in a continuous state space, how can it guarantee it will find the measure zero point? It seems the discussion about intuition is lacking here.

I was surprised to see no mention of 'Lévy flights', which seem distinct but related, and appear to provide a similar exploration heuristic in animals. It would be good to add a discussion about the similarities and differences to this.

After SGD I would include a citation to Robbins–Monro, as well as the Leon Bottou one.

In the related work section you should mention the use of Bayesian utility theory to guiding exploration, e.g., https://arxiv.org/abs/1807.09647, also some of part of this recent book draft may be relevant: http://www.cse.chalmers.se/~chrdimi/downloads/book.pdf

**Experience Assessment:**

I have published in this field for several years.

**Review Assessment: Checking Correctness Of Derivations And Theory:**

I assessed the sensibility of the derivations and theory.

**Review Assessment: Checking Correctness Of Experiments:**

I assessed the sensibility of the experiments.

**Review Assessment: Thoroughness In Paper Reading:**

I read the paper at least twice and used my best judgement in assessing the paper.

---

> ### Author Response · Authors · 2019-11-11
> **Response to Official Blind Review #1**
>
> We would like to thank Reviewer 1 for the thorough review and helpful suggestions.
>
>
> We now clarify in the paper that The proposed algorithm is most suitable to fully observable, continuous control, sparse-reward problems. It can work in other settings but would not necessarily be as efficient as RL algorithms tailored for these specific cases (e.g. Q-learning for discrete spaces).
>
> We note the concerns about extending the proposed algorithm to other RL problems. As mentioned in Section 6, Assumption 2 mostly restricts R3L in its basic form to simulation problems. Bridging the gap to real-world tasks can be done using sim-to-real techniques, which is left as future work in Section 6. For problems that don't come equipped with a goal set, one can be defined in terms of the return for purposes of the RRT, as shown in Theorem 2. For example, the pendulum task does not have any absorbing state, but we can define trajectories with a positive return as successful.
>
> As the problem domains become increasingly challenging, a larger tree will be needed to explore the trajectory space, and the memory requirements will grow. In the extreme case, the tree could be implemented in a databse in which nearest neighbour queries can be implemented efficiently (e.g. O(log(N)) with kd-tree, or O(1) with LSH). Further, we note that even the naive implementation was able to handle a considerable number of dimensions (78 state space and 20 action space dimensions) on a laptop with only 8GB of memory.
>
> "project $s_{rand}$ onto $\mathcal{F}_{goal}$". We actually directly sample  $s_{rand}$ from $\mathcal{F}_{goal}$. We have updated this in the algorithm.
>
> As mentioned in Section 3.1, the state space is treated as Euclidean. While this is a sufficient metric of nearness for the problems investigated in the paper, some problems may require a different metric (e.g. images).
>
> $\pi_l: \mathcal{S} \times \mathcal{S} \rightarrow \mathcal{A}$ is a function mapping a state and a goal state (a state difference in practice) to an action.
> It can be represented in a variety of ways, and its representation is not critical to the algorithm (indeed, it can be replaced with a random policy as shown in Table 1). As such, due to limited space in the paper, we deferred discussion to S3.2 in the supplement. We agree that the exact definition and flexibility in representation is worth mentioning in the paper body. When a new RL transition is available,  $\pi_l$ is updated with a new point of input $(s_{near}, s_{new})$ and target $a$. The specific update depends on the choice of model for $\pi_l$, e.g. SGD step for a neural network or an analytical update for a non-parametric model.
>
> Theoretical results do not rely on compactness, as the RRT will find a successful trajectory even if the goal set is open. However, the state space needs to be bounded, or else sampling from it is impractical. Further, If a goal set has measure zero, then the RRT will almost surely not find it, but the same is true for any RL algorithm.
>
> We agree with the suggestion to cite Robbins-Monro and to discuss Bayesian utility theory, and will add these to the paper.

---

### Official Review · AnonReviewer4 · 2019-10-31
**Official Blind Review #4**

**Rating:** 3

**Review:**

This paper proposes a method, R3L, for exploration in reinforcement learning. R3L performs an exploration procedure before policy optimization. In R3L exploration, it considers the task as a planning problem, and applies RRT to find feasible solutions (trajectories). Then it applies a warm start procedure for policy optimization, where the feasible solutions found by RRT are used to initialize the policy by supervised learning. The paper provides theoretical guarantees of R3L exploration finding feasible solutions. Empirically, the algorithmic contribution is demonstrated by comparing with information theoretical exploration methods in benchmark control domains.

The paper makes two strong assumptions, which are made to guarantee RRT can be successfully applied in the task. However, it is unlikely that these two assumptions can hold in RL problems we consider in general, where the learning agent only have access to the transition data gathered by interactions with the environment. This makes the proposed R3L algorithm not a general method for exploration in RL. Due to this reason I think this paper should be rejected.

The first assumption is that random states can be uniformly sampled from the MDP state space. The author further argue that sampling a random state is typically equivalent to trivially sampling a hyper-rectangle. But isn't this a domain-dependent property? For example, how to sample a random state in Atari games and decide if it is valid? By the method proposed in the paper, one need to sample a random image, then apply a function to decide if the image is valid in the game. How to get such a function? The second assumption is that the environment state can be set to an arbitrary state. This basically assumes the learning agent is available to the transition function, so that a new state can be added to the current search tree. But again, this assumption might not be appropriate in the general RL framework.

For the theoretical contribution, the paper shows that RRT is complete with high probability, which is a standard result of RRT. In experiments, R3L is compared with VIME. But is this a fair comparison since R3L assumes to have a generative model? The tested domains are picked such that RRT can be directly applied. Can R3L be applied in domains like mujoco or Atari games?

The main idea of this paper is to deal with exploration using planning algorithms. But once a generative model of the environment is given, the exploration problem will be very different with the exploration considered in RL. I would like to improve my score if the author can demonstrate the efficiency of R3L with a learned generative model.

Minor issues:

1. Can you give more details about how pi_l is learned in Algorithm 1? In line 9 an action is generated using pi_l(s_near, s_rand-s_near). But in line 11, the action is again updated by ({s_near, s_rand-s_near}, a), which is very confusing.

2. The notations for state and valid state are very confusing. In 3.1, the transition and reward functions are defined on S. But later in the paper, S contains states that are not valid. What's the transitions and rewards on invalid states?


**Experience Assessment:**

I have published one or two papers in this area.

**Review Assessment: Checking Correctness Of Derivations And Theory:**

I carefully checked the derivations and theory.

**Review Assessment: Checking Correctness Of Experiments:**

I assessed the sensibility of the experiments.

**Review Assessment: Thoroughness In Paper Reading:**

I read the paper at least twice and used my best judgement in assessing the paper.

---

> ### Author Response · Authors · 2019-11-11
> **Response to Official Blind Review #4**
>
> We would like to thank Reviewer 4 for the thorough review and helpful suggestions.
>
> Reviewer 4 has raised some concerns about the assumptions. First, it is important to note that these assumptions only pertain to the demonstration generation (planning) phase, not the RL phase. We will clarify this in the text.
>
> Regarding Assumption 1, we stress that the random states $s_{rand}$ sampled from $\mathcal{S}$ do not need to be valid, as stated in the paper following Assumption 1. $s_{rand}$ is not added to the tree. Hence, there is no need for a function that will test states for validity. Sampling a (not necessarily valid) state from $\mathcal{S}$ is indeed domain specific, as Reviewer 4 notes. Nonetheless, it reduces to sampling a hyper-rectangle in all common RL benchmark tasks (OpenAI Gym, MuJoCo, Atari Learning Environment, DMControl), since all of these tasks define the state space as a hyper-rectangle.
>
> We have changed Assumption 2 to make it milder and more accurate. It now reads: "The environment state can be set to a previously visited state $s \in \mathbb{T}$". In simulation, setting the environment to a specific state can be implemented as a variable assignment. As such, this step does not involve a transition from the current state to the target state, and does not require access to the transition dynamics, and does not involve a generative model. As discussed in Section 6, policies learned in simulated environments can be transferred to real world tasks using sim-to-real techniques, which is left as future work. We will clarify the connection between simulation and Assumption 2 in the main text.
>
> As noted above, $s_{rand}$ is not added to the tree. $s_{new}$ (almost always different from $s_{rand}$) is generated by executing action $a=\pi_l(s_{near}, s_{rand}-s_{near})$ in state $s_{near}$. Assumption 2 is used to set the state to $s_{near}$ for this purpose. Action $a$ attempts to reach $s_{rand}$ but is not guaranteed to succeed. This is a classic RL transition $(s_{near}, a, s_{new})$, which does not require knowing transition dynamics, and enforces that $s_{new}$ is a valid state.
>
> Regarding the theoretical contribution, our results extend RRT planning guarantees to the setting of MDPs, where the goal might be specified not as a subset of the state space but in terms of the return (cf. Theorem 2). Further, Theorem 3 is not a standard result in RRTs or in RL.
>
> As to mujoco and Atari environments, we note that three of the domains considered in the paper (Reacher, Fetch Reach and Hand Reach) are mujoco domains.
> R3L is most suitable to fully observable, continuous control, sparse-reward problems. R3L should be applicable as-is to Atari problems, but will not necessarily be as efficient as methods tailored for discrete state and action spaces, or for image data. Extending R3L to make it practical on high-dimensional tasks is mentioned as future work in Section 6. We will clarify that this includes Atari/image data problems.
>
> Concerning the minor issues:
>   * The local policy is defined as $\pi_l: \mathcal{S} \times \mathcal{S} \rightarrow \mathcal{A}$, mapping a state and a goal state to an action. line 11 does not update $a$, but rather the policy model $\pi_l$. The action $a$ is used as supervision for this update.
>   * An invalid state is e.g. a robotic arm being inside a wall. Such a state is implausible, but in practice most MDPs include it in their definition of the state space. Since this paper is in the sparse reward setting, the reward for such a state would be $-1$ should it somehow be encountered. The transition dynamics in that state would depend on the MDP. In the example of the robotic arm getting stuck inside a wall, this would be an absorbing state, as no action can change the pose of the arm.

---

### Decision · Program_Chairs · 2019-12-19

**Decision:**

Reject

**Comment:**

This was a borderline paper, with both pros and cons.  In the end, it was not considered sufficiently mature to accept in its current form.  The reviewers all criticized the assumptions needed, and lamented the lack of clarity around the distinction between reinforcement learning and planning.  The paper requires a clearer contribution, based on a stronger justification of the approach and weakening of the assumptions.  The submitted comments should be able to help the authors strengthen this work.